

# Belowground fungal community diversity, composition and ecological functionality associated with winter wheat in conventional and organic agricultural systems

Sigisfredo Garnica[1], Ronja Rosenstein[2] and Max Emil Schön[3]

[1] Instituto de Bioquímica y Microbiología, Facultad de Ciencias, Universidad Austral de Chile, Valdivia, Isla Teja, Chile
[2] Institute of Evolution and Ecology, Plant Evolutionary Ecology, University of Tuebingen, Tuebingen, Germany
[3] Department of Cell and Molecular Biology, Science for Life Laboratory, Uppsala University, Uppsala, Sweden

Corresponding author
Sigisfredo Garnica,
sigisfredo.garnica@uach.cl

## ABSTRACT

Understanding the impacts of agricultural practices on belowground fungal communities is crucial in order to preserve biological diversity in agricultural soils and enhance their role in agroecosystem functioning. Although fungal communities are widely distributed, relatively few studies have correlated agricultural production practices. We investigated the diversity, composition and ecological functionality of fungal communities in roots of winter wheat (*Triticum aestivum*) growing in conventional and organic farming systems. Direct and nested polymerase chain reaction (PCR) amplifications spanning the internal transcribed spacer (ITS) region of the rDNA from pooled fine root samples were performed with two different sets of fungal specific primers. Fungal identification was carried out through similarity searches against validated reference sequences (RefSeq). The R package 'picante' and FUNGuild were used to analyse fungal community composition and trophic mode, respectively. Either by direct or cloning sequencing, 130 complete ITS sequences were clustered into 39 operational taxonomic units (OTUs) (25 singletons), belonging to the Ascomycota (24), the Basidiomycota (14) and to the Glomeromycota (1). Fungal communities from conventional farming sites are phylogenetically more related than expected by chance. Constrained ordination analysis identified total N, total S and Pcal that had a significant effect on the OTU's abundance and distribution, and a further correlation with the diversity of the co-occurring vegetation could be hypothesised. The functional predictions based on FUNGuild suggested that conventional farming increased the presence of plant pathogenic fungi compared with organic farming. Based on diversity, OTU distribution, nutrition mode and the significant phylogenetic clustering of fungal communities, this study shows that fungal communities differ across sampling sites, depending on agricultural practices. Although it is not fully clear which factors determine the fungal communities, our findings suggest that organic farming systems have a positive effect on fungal communities in winter wheat crops.

## INTRODUCTION

Crop production has increased over time because of the growing global demand particularly for food, feed and fuel supplies. To meet this demand, diverse agricultural farming systems are being developed in an attempt to contribute to increased crop production yields (*Tilman et al., 2011*). Although there are many different agricultural techniques, these can be generalised, depending on the techniques used, as either conventional or organic (sustainable) systems. Basically, conventional farming uses synthetic chemicals and fertilisers and is designed specifically to generate maximal yields, whereas organic farming aims to produce a number of crops without the use of synthetic chemicals or fertilisers (*Gomiero, Pimentel & Paolett, 2011*). The effects of such methods on physical and chemical soil properties have been well studied. Although previous studies have demonstrated that crop and soil management practices have a significant impact on soil microorganisms (*Seghers et al., 2004*; *Zarb et al., 2005*), we still have little knowledge of how land use may affect the abundance and composition of specific taxonomic groups.

Some previous studies have found greater soil fungi abundance (*Shannon, Sen & Johnson, 2002*; *Tautges et al., 2016*) and higher arbuscular mycorrhizal fungal colonisation (*Gryndler et al., 2006*; *Mäder et al., 2002*; *Oehl et al., 2004*) in organic than in conventionally managed wheat (*Triticum aestivum*) production systems. Perhaps the fungal groups of major interest are probably those that live in endophytic association with crop plants. For winter wheat, one of the most important food crop plants worldwide (*O'Hanlon et al., 2012*), previous studies by *Carter et al. (1999)*, *Larran et al. (2007)*, *Riess et al. (2014)* and *Verbruggen et al. (2014)* have examined the effects of agricultural management on the diversity and composition of root fungal endophyte communities. By using a combination of cultural characteristics and subsequent identification by sequencing rDNA, *Carter et al. (1999)* examined fungal endophytes associated with plant roots. *Larran et al. (2007)* used classical cultivation and morphology-based methods to identify endophytic fungi isolated from aboveground plant tissues of different cultivars. By using cultivation and sequencing methods, *Riess et al. (2014)* focused on the detection of endophytic Sebacinales from roots across different agricultural habitats. *Verbruggen et al. (2014)* found that Sebacinales but not total roots associated fungal communities are affected by land use intensity. In all these studies, fungal molecular operational taxonomic units (OTUs) or species, mainly Ascomycota and some Basidiomycota, were detected to be the most common root-colonising endophytes. So far, some experimental studies for agriculturally relevant crop plants have demonstrated that endophytes can increase tolerance to environmental factors like drought and heat, as reported by *Hubbard, Germida & Vujanovic (2014)*, or provide resistance against herbivory (*Gurulingappa et al., 2010*). They have also been shown to be involved in fungal pathogen protection and/or growth promotion in wheat (*Rabiey, Ullah & Shaw, 2015*). However, apart from these

potential effects on plants, we still have poor knowledge of how agricultural practices may influence these root-colonising microbial communities in the field.

In this work, we selected five sites each from the two most commonly applied farming systems to examine specifically how belowground fungal communities respond to different agricultural practices using winter wheat as the target plant, which is of great economic importance and is the predominant crop in Germany (*Statistisches Bundesamt Wiesbaden, 2015*). The main proportion of cereal cropland (96% in 2013) in Germany is based on conventional agriculture; however, between 2003 and 2013, the number of agricultural holdings practicing organic farming increased by 30% (*Statistisches Bundesamt Wiesbaden, 2014*). We sequenced the internal transcribed spacer (ITS) region of the rDNA directly or after cloning and analysed it to identify and characterise the belowground fungal communities associated with roots of winter wheat from agricultural sites under conventional or organic management. Four specific questions are addressed: (i) What is the diversity of these communities associated with winter wheat associated with these agricultural practices? (ii) What are the patterns in phylogenetic structure characterising such communities? (iii) Do biotic and/or abiotic environmental factors affect the phylogenetic diversity and structure of the communities associated with winter wheat? (iv) What is the ecological function of these fungal communities? In order to cover as much of the fungal diversity as possible, we used two different sets of specific primers to characterise the diversity in the roots. We used this sequence-based approach to characterise the diversity and community composition of these microorganisms across sites under two different agricultural practices and associated with the influence of selected biotic and/or abiotic environmental factors.

## MATERIALS AND METHODS

### Study sites

During the end of April 2015, winter wheat (*Triticum aestivum*) plants were sampled from 10 different sites located in the surroundings of Tübingen, Baden-Württemberg, Germany (see Fig. 1). We sampled agricultural fields treated under conventional farming management located in Pfrondorf (C1, C2 and C4) and Entringen (C3 and C5) as well as fields under organic farming management located in Unterjesingen (O1, O2, O3, O4 and O5). Within each agricultural field, plants were collected along a diagonal transect. Organic farming sites most commonly featured frequent and diverse co-occurring herbaceous plants. Site O2 was also surrounded by shrubs and site O3 was located next to a group of ectomycorrhizal (ECM) trees. For the conventional farming sites, sites C2 and C3 had more diverse co-occurring plant species. A complete list of co-inhabiting plants at each collection site is given as Supplemental Information (Table S1). The farmers Ulrich Bechtle (Unterjesingen) and Eckhart Wizemann (Entringen, Pfrondorf) allowed us to collect plant specimens from their crop fields.

### Plant and soil sample collection

Plants from a sampling transect, defined as the longest diagonal between two of the corners of each site, were sampled. Along each transect, 10 plots (labelled A–J) were positioned

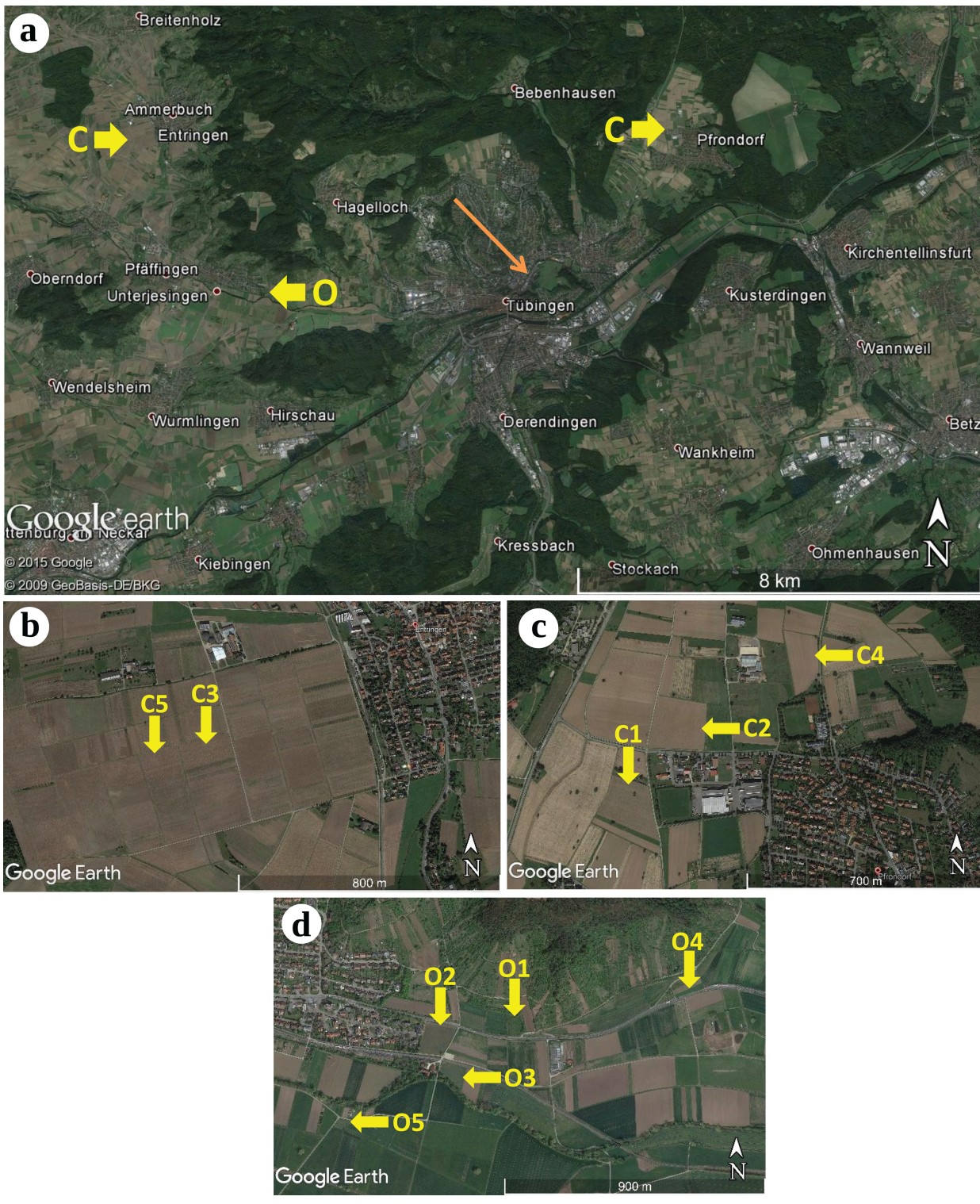

**Figure 1 Geographic locations of the sampling sites of winter wheat (*Triticum aestivum*) around Tübingen (Germany), including two conventional farming sites.** (A) Overview: conventional farming (C) Entringen (B) and Pfrondorf (C) and organic farming (O) in Unterjesingen (D) (Map data: © 2015 Google, © 2009 GeoBasis-DE/BKG). Detailed view of the sampling sites: (B) two sites in Entringen (Map Data: Google Earth), (C) three sites in Pfrondorf (Map Data: Google Earth), and (D) five sites in Unterjesingen (Map Data: Google Earth) respectively.

at similar distance intervals, with the two outer plots (A and J) being placed directly in the corners, the distance between plots ranged from 8 m to 28 m (Table S1). Within each sampling plot, at least five different plants at the tillering stage were randomly chosen and carefully dug out of the ground with a shovel, retaining as much of the root system as possible and a portion of soil. Plant size ranged from approximately 15 cm to 30 cm; individuals growing in the corners of the sites were always smaller than those growing in the central area. Sampled plants were stored in plastic bags at 4 °C until further processing.

The root systems of each plant were soaked in tap water to loosen the soil and were then washed extensively under running tap water. Subsequently, the five plants per plot with the highest number of healthy roots were chosen for molecular analysis. The roots were washed carefully under running purified water, then three fine roots per plant were randomly selected, cut off with a sterile scalpel and forceps, and were placed into a Petri dish filled with double-distilled and sterilised water. Root surface sterilisation was performed via a modified version of the method of *Sun et al. (2008)*, which consisted of rinsing the roots in ethanol (70%) for 5 min in a Petri dish, then in sodium hypochlorite (NaOCl 1%) for another 5 min and finally washing them with double-distilled water three times to remove any remaining chemicals. After that procedure, all 15 root portions from each plot (three root segments from five plants each) were pooled together as one sample, placed into 2-mL Eppendorf tubes and dried at 50 °C in a Dörrex device for several days. In total, the roots of 500 plants were randomly sampled (representing 100 pooled samples), 250 plants for each of the two different farming methods. Herbarised plants were deposited in the Herbarium Tubingense (TUB) as Nos. TUB 021591 to TUB 021628.

In addition, from each plot, portions of soil adhering to the roots of the sampled plants were removed and stored in paper bags for drying. Two soil samples per site were analysed independently: soil from the two edge plots was mixed and pooled as one sample, whereas the soil from the remaining middle plots was used as a second sample. Measurements of total carbon and nitrogen content as well as organic carbon content, pH and plant-available phosphorus content were conducted at the Soil Science and Geoecology Laboratory of the University of Tübingen. Additionally, the carbon-to-nitrogen ratio was calculated and the humus content was estimated by multiplying the organic carbon content by 1.72 as described by Ad-hoc Arbeitsgruppe *Boden (2005)*.

## DNA extraction, amplification, cloning and DNA sequencing

Dried fine root pools were homogenised with a mixer mill (MM 300; Retsch GmbH, Haan, Germany) with two steel beads per Eppendorf tube at a frequency of 30 Hz for 10 min. Total gDNA was extracted with the InnuPREP Plant DNA Kit (Analytik Jena AG, Jena, Germany) following the manufacturer's protocol, except that the amounts of lysis buffer, RNase, protein kinase and binding solution were doubled in order to dissolve the large volume of homogenised root material completely. To detect root-associated fungi, PCRs were performed to amplify the ITS region by following two different approaches: (i) the first approach was used to amplify the ITS region including the 5.8S domain with the fungal-specific primer ITS1F (*Gardes & Bruns, 1993*) and the universal

primer ITS4 (*White et al., 1990*); (ii) in the second approach, the ITS region and the large subunit domain D1/D2 were amplified with the fungal-specific primers NSI1 (*Martin & Rygiewicz, 2005*) and NL4 (*White et al., 1990*). PCR products from the second approach (for samples producing negative or weak PCR products) were subsequently used for nested PCRs with the primer combination ITS1F–ITS4. In both cases, MangoTaq™ DNA Polymerase (Bioline GmbH, Lückenwalde, Germany) was used for amplification and the reactions consisted of a volume of 25 μL containing 5 μL coloured reaction buffer, 14.5 μL double-distilled $H_2O$, 0.75 μL $MgCl_2$ (50 mM), 1 μL dNTPs (5 mM), 0.5 μL of the forward primer (25 pmol/μL), 0.5 μL of the reverse primer (25 pmol/μL), 0.25 μL MangoTaq DNA Polymerase (2 unit/μL) and 2.5 μL of the DNA template. For direct PCRs with the NSI1–NL4 primers, PCR reactions consisted of half the volume. PCRs were run with the thermo programme TOUCH 60 under the following conditions: initial denaturation at 94 °C for 3 min; 10 cycles each of denaturation at 94 °C for 30 s, annealing at 60 °C for 45 s, with each cycle decreasing by 1 °C, and elongation at 72 °C for 1 min 15 s, followed by 26 cycles each of denaturation at 94 °C for 30 s, annealing at 50 °C for 45 s, elongation at 72 °C for 1 min 15 s and a final extension phase at 72 °C for 7 min. PCR products were verified by running an agarose gel electrophoresis with 0.7% agarose and 1× Tris-Borat-EDTA (TBE) buffer for 15 min at 143 V. The PCR products were subsequently stained with Midori Green (NIPPON Genetics EUROPE GmbH, Dueren, Germany) for 30 min and visualised under UV illumination.

Positive PCR products that could not be sequenced directly were cloned into competent *Escherichia coli* cells with the TOPO TA Cloning® Kit for Sequencing (Invitrogen, Life Technologies GmbH, Darmstadt, Germany). The cloning reaction for one sample contained 0.33 μL of a salt solution, 0.33 μL double distilled $H_2O$, 0.33 μL of pCR®4-TOPO vector (Invitrogen, Carlsbad, CA, USA) and 1 μL of the undiluted PCR product. Samples were incubated at room temperature for 30 min. Afterwards, 19 μL of One Shot® TOP10 chemically competent *E. coli* cells (Invitrogen, Carlsbad, CA, USA) was carefully added to the cloning reaction and the mixture subsequently incubated on ice for 30 min. Heat shock was conducted in a water bath at 42 °C for 30 s, after which the cells were kept on ice for 2 min. Finally, after adding 80 μL of Super Optimal broth Catabolite repression (S.O.C.) medium, the mixture was placed into an incubator at 37 °C and 200 rpm for 1, then plated onto Lysogeny Broth (LB)-medium including Kanamycin (1 mL/L). Plates were incubated at 37 °C overnight. Up to 12 bacterial colonies per sample were picked and directly used for PCRs with the forward and reverse primers M13F/M13R (Invitrogen, Carlsbad, CA, USA) and MangoTaq DNA Polymerase in 25-μL reactions. Amplification in the thermo programme M13 consisted of initial denaturation at 94 °C for 7 min, followed by 32 cycles each of denaturation at 94 °C for 20 s, annealing at 55 °C for 40 s, elongation at 72 °C for 2 min 40 s and a final extension at 72 °C for 7 min. For samples with a large number of positive PCR products, eight clones were digested enzymatically with *Hinf1* (New England BioLabs GmbH, Frankfurt am Main, Germany) to detect restriction fragment length polymorphisms. To check for different restriction patterns, gel electrophoresis was run with 1% agarose and 1× Tris-EDTA (TE)

buffer at 120 V for 40 min, followed by staining with Midori Green (Nippon Genetics EUROPE GmbH, Dueren, Germany) for 30 min and visualisation under UV illumination. All PCR products showing different restriction patterns were sequenced, as well as some with the same pattern.

Amplified PCR products were cleaned up with diluted (1:20) ExoSAP-IT® (Affymetrix UK Ltd., High Wycombe, UK) following the manufacturer's protocol. Cycle sequencing was performed on 4 μL of a 1:6 diluted ABI PRISM BigDye® Terminator v3.1 Cycle Sequencing Kit (Applied Biosystems, Life Technologies GmbH, Darmstadt, Germany) and 1 μL of the primers ITS1F–ITS4 and, if necessary, 5.8SR (*Vilgalys & Hester, 1990*), ITS2 or ITS1 (*White et al., 1990*) (12 pmol/μL) for directly sequenced PCR products and the primers M13F–M13R (12 pmol/μL) for cloned PCR products. The thermo programme ABI31CYCL was used for amplification under following conditions: initial denaturation at 96 °C for 1 min, followed by 27 cycles each of denaturation at 96 °C for 10 s, annealing at 50 °C for 5 s and elongation at 60 °C for 4 min. Afterwards, the cycle sequencing products were precipitated in 75% isopropanol, washed with 80% ethanol and dried in a vacuum centrifuge for approximately 5 min. DNA pellets were dissolved in 15 μL Hi-Di™ formamide (Applied Biosystems, Life Technologies, Warrington, UK) for 10 min and sequencing was conducted on an ABI3130xl Genetic Analyzer (Applied Biosystems, Foster City, CA, USA). A list of primers used in this study, including their sequences, is given in the Supplemental Information (Table S2).

## Sequence editing, chimaera checking and BLAST search

Forward and reverse ITS sequences were assembled automatically and edited manually by Sequencher v.4.9 (Gene Codes Corporation, Ann Arbor, MI, USA).

All sequences were first checked for PCR chimaeras with VSEARCH v2.4.3 (*Rognes et al., 2016*), implementing the UCHIME algorithm (*Edgar et al., 2011*). All chimeric and 'borderline' chimeric sequences were excluded from further analyses. Non-chimeric ITS sequences with GenBank accession nos. KY430446 to KY430584 were used as queries in a search against the GenBank ITS RefSeq Database (BioProject PRJNA177353, accessed 12 December 2018) with the programme MALT (*Herbig et al., 2016*), using the default parameters. The ITS sequences generated in this study were then annotated with the most specific taxonomic label inferred.

## Fungal diversity estimation

Fungal ITS sequences were clustered into OTUs based on a ≥97% similarity threshold with the UCLUST algorithm implemented in VSEARCH (*Edgar, 2010*; *Rognes et al., 2016*). We estimated the alpha diversity of those samples by calculating Hill numbers in the R package 'vegan' (*Oksanen et al., 2016*). We also computed the number of OTUs that were shared by the two farming systems and the number of OTUs that were unique to a specific system. A pairwise species distance matrix was computed by first calculating the pairwise sequence distances via the alignment-free approach of spaced-words frequencies (*Leimeister et al., 2014*). Pairwise sequence distances were subsequently averaged to pairwise species (OTU) distances.

## Phylogenetic community structure

We calculated the standardised effect sizes of the mean phylogenetic distance (MPD, equivalent to—net relatedness index (NRI), *Webb et al., 2002*) and the mean nearest phylogenetic taxon distance (MNTD, equivalent to—nearest taxon index (NTI)) metrics as implemented in the R package 'picante' to assess the composition of fungal samples in conventional vs. organically managed farming sites (*Kembel et al., 2010*). We compared the MPD and MNTD values for our samples with random samples generated under a null model that shuffled species labels randomly across the complete phylogeny.

## Differences in environmental factors between farming methods

The differences in measured soil parameters between organic and conventional plots were visualised via non-metric multidimensional scaling (NMDS) using the Bray-Curtis index with functions implemented in the 'vegan' package (*Oksanen et al., 2016*) for R (*R Development Core Team, 2011*). Specifically, NMDS was performed to analyse the effect of, for example plant-available phosphorus, carbon-to-nitrogen ratio, pH value (pH $CaCl_2$) and humus content (Table S3).

We conducted a transformation-based Redundancy Analysis (tb-RDA) to assess the effect of the environmental parameters on the OTU distribution (*Legendre & Legendre, 2012*). Species abundances were transformed using Hellinger transformation (*Legendre & Gallagher, 2001*) and variables were selected using backward selection using functions from the 'vegan' package. Additionally, a PERMANOVA was conducted (using the Bray–Curtis index) to test whether the farming system has a significant influence on the OTU distribution (*Anderson, 2001*).

## Functional guilds

We classified the identified taxa into ecological guilds with the FUNGuild tool and database (the database is available at www.stbates.org/funguild_db.php, accessed 23 January 2020) (*Nguyen et al., 2016*), to determine if specific functional groups of fungi (e.g. pathotrophs, saprotrophs, symbiotrophs, etc.) differed between sites and/or the farming methods applied. All OTUs that did not match any taxa in the database were categorised as 'unassigned'.

# RESULTS

## PCR success and belowground fungal diversity

A total of 54% of the pooled root samples yielded PCR products; 39 samples resulted positive with the primer set ITS1F/ITS4 and 22 with NSI1/NL4, respectively (Table S2). Seven samples produced positive results with both sets of primers. Nested PCRs from the primer set NSI1–NL4 yielded strong bands but they were not directly sequenceable and, after cloning, the PCR produced only very few colonies. In total, 130 full-length non-chimeric ITS sequences clustered in 39 OTUs, 23 of which resulted in unique genotypes. In six cases, identical sequences for the same sample were obtained from both primer sets.

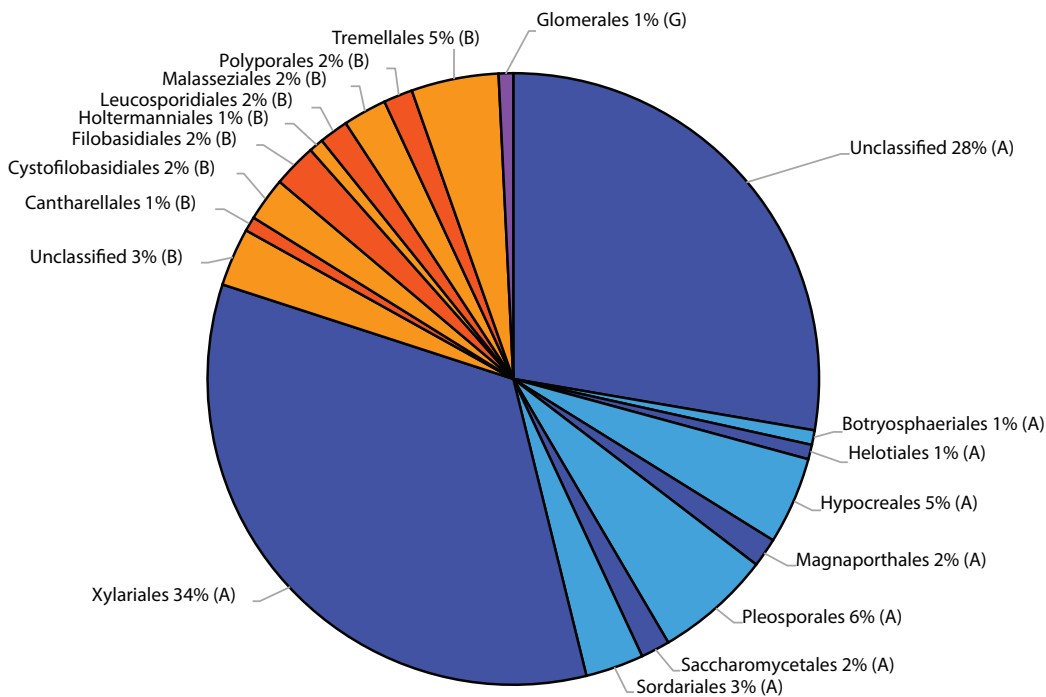

**Figure 2 Taxonomic distribution at order level of fungal sequences associated with roots of winter wheat (*Triticum aestivum*) based on similarity searches against the fungal ResSeq database.** (A) Ascomycota, (B) Basidiomycota and (G) Glomeromycota.

Belowground fungal communities were dominated by Ascomycota with 104 sequences (80%) and Basidiomycota with 25 sequences (19.23%), whereas the phylum Glomeromycota with one sequence (0.77%) was less frequent (Fig. 2). When we clustered these sequences into OTUs, 24 were classified as Ascomycota, 14 as Basidiomycota and one as Glomeromycota. Within the Ascomycota, the OTUs #20 (39 sequences), #7 (30 sequences) and #26 (five sequences) contained the most sequences and in the OTUs with the most sequences were the Basidiomycota #35 (four sequences), #5, #8 and #0 (three sequences each). The most frequent OTUs, #20 and #7, were detected in seven and eight different samples from conventionally and organically managed sites, respectively. The remaining OTUs were found either in a single sample or in up to four different samples. Up to 12 different fungal sequences (#20 on C2) from the same OTU were generated from the same sample. Within each farming system, some OTUs were found in samples along the entire transect. The OTUs show a nearly linear accumulation and the OTU sampling did not reach saturation (Table 1). Out of all OTUs detected, 10 occurred exclusively in conventional farms (three non-singletons) and 19 in organic farms (one non-singleton), whereas 10 OTUs were present in both farming systems. Based on a comparison with reference sequences, several OTUs have broadly disjunct distributions, either continental or locally restricted. The diversity indices of the belowground fungal communities are given in Fig. 3.

**Table 1 Taxonomic identifications based on similarity searches against the fungal RefSeq database for fungal sequences associated with the roots of winter wheat (*Triticum aestivum*) from sites under organic and conventional management.**

| Sequence ID | MOTU | Species | Trophic mode | Farming system |
|---|---|---|---|---|
| KY430446 | 7 | Helotiales incertae sedis | – | Conventional |
| KY430447 | 7 | Ascomycota | – | Organic |
| KY430448 | 7 | Ascomycota | – | Organic |
| KY430449 | 7 | Ascomycota | – | Organic |
| KY430451 | 19 | Schizothecium | Saprotroph | Conventional |
| KY430452 | 38 | *Candida sake* | Saprotroph | Organic |
| KY430453 | 38 | *Candida sake* | Saprotroph | Organic |
| KY430454 | 27 | Pleosporineae | – | Conventional |
| KY430455 | 29 | Didymellaceae | Pathotroph-Saprotroph | Conventional |
| KY430456 | 28 | Pleosporineae | – | Organic |
| KY430457 | 20 | Microdochium | Pathotroph-Symbiotroph | Organic |
| KY430458 | 20 | Microdochium | Pathotroph-Symbiotroph | Conventional |
| KY430459 | 20 | Microdochium | Pathotroph-Symbiotroph | Conventional |
| KY430460 | 20 | Microdochium | Pathotroph-Symbiotroph | Conventional |
| KY430461 | 20 | Microdochium | Pathotroph-Symbiotroph | Conventional |
| KY430462 | 20 | Microdochium | Pathotroph-Symbiotroph | Organic |
| KY430463 | 35 | Vishniacozyma victoriae | Pathotroph-Saprotroph-Symbiotroph | Organic |
| KY430465 | 6 | Agaricomycetes | – | Organic |
| KY430466 | 35 | Vishniacozyma victoriae | Pathotroph-Saprotroph-Symbiotroph | Conventional |
| KY430467 | 5 | Filobasidium | Saprotroph | Conventional |
| KY430468 | 5 | Filobasidium | Saprotroph | Organic |
| KY430469 | 20 | Microdochium | Pathotroph-Symbiotroph | Conventional |
| KY430470 | 20 | Microdochium | Pathotroph-Symbiotroph | Conventional |
| KY430471 | 7 | Ascomycota | – | Organic |
| KY430472 | 31 | Darksidea | Symbiotroph | Organic |
| KY430473 | 26 | Fusarium equiseti | Pathotroph-Saprotroph-Symbiotroph | Conventional |
| KY430474 | 36 | Ascomycota | – | Organic |
| KY430475 | 20 | Microdochium | Pathotroph-Symbiotroph | Organic |
| KY430476 | 26 | Fusarium equiseti | Pathotroph-Saprotroph-Symbiotroph | Conventional |
| KY430477 | 26 | Fusarium equiseti | Pathotroph-Saprotroph-Symbiotroph | Conventional |
| KY430478 | 33 | Podospora | Saprotroph-Symbiotroph | Organic |
| KY430479 | 8 | *Cystofilobasidium macerans* | Saprotroph | Organic |
| KY430480 | 20 | Microdochium | Pathotroph-Symbiotroph | Conventional |
| KY430481 | 4 | Polyporales | – | Organic |
| KY430482 | 11 | Botryosphaeriales | – | Organic |
| KY430483 | 13 | Lentitheciaceae | – | Organic |
| KY430484 | 5 | Filobasidium | Saprotroph | Organic |
| KY430485 | 7 | Ascomycota | – | Organic |
| KY430487 | 24 | Leotiomycetes | – | Organic |

 

| Sequence ID | MOTU | Species | Trophic mode | Farming system |
|---|---|---|---|---|
| **Table 1 (continued)** | | | | |
| KY430488 | 7 | Ascomycota | – | Organic |
| KY430489 | 30 | Ascomycota | – | Organic |
| KY430490 | 15 | *Helminthosporium velutinum* | Pathotroph | Organic |
| KY430491 | 34 | Vishniacozyma tephrensis | Pathotroph-Saprotroph-Symbiotroph | Organic |
| KY430492 | 7 | Ascomycota | – | Conventional |
| KY430493 | 20 | Microdochium | Pathotroph-Symbiotroph | Conventional |
| KY430494 | 20 | Microdochium | Pathotroph-Symbiotroph | Organic |
| KY430496 | 7 | Ascomycota | – | Organic |
| KY430497 | 20 | Microdochium | Pathotroph-Symbiotroph | Conventional |
| KY430498 | 14 | *Funneliformis mosseae* | Symbiotroph | Organic |
| KY430499 | 7 | Ascomycota | – | Organic |
| KY430500 | 35 | Vishniacozyma victoriae | Pathotroph-Saprotroph-Symbiotroph | Conventional |
| KY430501 | 7 | Ascomycota | – | Conventional |
| KY430502 | 7 | Ascomycota | – | Organic |
| KY430503 | 20 | Microdochium | Pathotroph-Symbiotroph | Conventional |
| KY430504 | 7 | Ascomycota | – | Organic |
| KY430505 | 20 | Microdochium | Pathotroph-Symbiotroph | Conventional |
| KY430506 | 20 | Microdochium | Pathotroph-Symbiotroph | Conventional |
| KY430507 | 20 | Microdochium | Pathotroph-Symbiotroph | Conventional |
| KY430508 | 20 | Microdochium | Pathotroph-Symbiotroph | Conventional |
| KY430509 | 12 | Slopeiomyces cylindrosporus | Pathotroph | Organic |
| KY430510 | 22 | Microdochium | Pathotroph-Symbiotroph | Organic |
| KY430511 | 22 | Microdochium | Pathotroph-Symbiotroph | Organic |
| KY430512 | 19 | Schizothecium | Saprotroph | Conventional |
| KY430514 | 20 | Microdochium | Pathotroph-Symbiotroph | Organic |
| KY430515 | 9 | *Leucosporidium golubevii* | – | Organic |
| KY430516 | 20 | Microdochium | Pathotroph-Symbiotroph | Organic |
| KY430517 | 21 | Periconia | Pathotroph-Saprotroph-Symbiotroph | Organic |
| KY430518 | 7 | Ascomycota | – | Organic |
| KY430519 | 12 | Slopeiomyces cylindrosporus | Pathotroph | Organic |
| KY430520 | 20 | Microdochium | Pathotroph-Symbiotroph | Organic |
| KY430521 | 20 | Microdochium | Pathotroph-Symbiotroph | Organic |
| KY430522 | 20 | Microdochium | Pathotroph-Symbiotroph | Conventional |
| KY430523 | 20 | Microdochium | Pathotroph-Symbiotroph | Conventional |
| KY430524 | 20 | Microdochium | Pathotroph-Symbiotroph | Conventional |
| KY430525 | 20 | Microdochium | Pathotroph-Symbiotroph | Conventional |
| KY430526 | 20 | Microdochium | Pathotroph-Symbiotroph | Conventional |
| KY430527 | 3 | Clavulina | Symbiotroph | Organic |
| KY430528 | 7 | Ascomycota | – | Organic |

(Continued)

| Table 1 (continued) | | | | |
|---|---|---|---|---|
| Sequence ID | MOTU | Species | Trophic mode | Farming system |
| KY430529 | 7 | Ascomycota | – | Organic |
| KY430530 | 10 | Leucosporidium | – | Organic |
| KY430531 | 26 | Fusarium equiseti | Pathotroph-Saprotroph-Symbiotroph | Conventional |
| KY430532 | 32 | *Podospora dimorpha* | Saprotroph-Symbiotroph | Organic |
| KY430533 | 26 | Fusarium equiseti | Pathotroph-Saprotroph-Symbiotroph | Conventional |
| KY430534 | 20 | Microdochium | Pathotroph-Symbiotroph | Conventional |
| KY430536 | 8 | *Cystofilobasidium macerans* | Saprotroph | Organic |
| KY430537 | 7 | Ascomycota | – | Organic |
| KY430539 | 7 | Ascomycota | – | Conventional |
| KY430540 | 22 | Microdochium | Pathotroph-Symbiotroph | Conventional |
| KY430541 | 20 | Microdochium | Pathotroph-Symbiotroph | Conventional |
| KY430542 | 7 | Ascomycota | – | Organic |
| KY430543 | 7 | Ascomycota | – | Organic |
| KY430544 | 7 | Ascomycota | – | Organic |
| KY430545 | 7 | Ascomycota | – | Organic |
| KY430546 | 7 | Ascomycota | – | Organic |
| KY430547 | 20 | Microdochium | Pathotroph-Symbiotroph | Conventional |
| KY430548 | 1 | *Malassezia restricta* | Pathotroph-Saprotroph | Conventional |
| KY430549 | 20 | Microdochium | Pathotroph-Symbiotroph | Conventional |
| KY430550 | 8 | *Cystofilobasidium macerans* | Saprotroph | Conventional |
| KY430551 | 20 | Microdochium | Pathotroph-Symbiotroph | Conventional |
| KY430552 | 25 | Ascomycota | – | Organic |
| KY430553 | 2 | *Malassezia restricta* | Pathotroph-Saprotroph | Conventional |
| KY430555 | 7 | Ascomycota | – | Organic |
| KY430556 | 22 | Microdochium | Pathotroph-Symbiotroph | Conventional |
| KY430557 | 22 | Microdochium | Pathotroph-Symbiotroph | Conventional |
| KY430558 | 7 | Ascomycota | – | Conventional |
| KY430559 | 7 | Ascomycota | – | Conventional |
| KY430560 | 4 | Polyporales | – | Conventional |
| KY430561 | 23 | Leotiomycetes | – | Conventional |
| KY430563 | 20 | Microdochium | Pathotroph-Symbiotroph | Conventional |
| KY430564 | 20 | Microdochium | Pathotroph-Symbiotroph | Conventional |
| KY430565 | 7 | Ascomycota | – | Conventional |
| KY430566 | 7 | Ascomycota | – | Conventional |
| KY430567 | 7 | Ascomycota | – | Conventional |
| KY430568 | 7 | Ascomycota | – | Conventional |
| KY430569 | 24 | Leotiomycetes | – | Conventional |
| KY430570 | 20 | Microdochium | Pathotroph-Symbiotroph | Conventional |
| KY430571 | 37 | *Dioszegia hungarica* | Pathotroph-Saprotroph-Symbiotroph | Conventional |

| Table 1 (continued) | | | | |
|---|---|---|---|---|
| Sequence ID | MOTU | Species | Trophic mode | Farming system |
| KY430572 | 20 | Microdochium | Pathotroph-Symbiotroph | Conventional |
| KY430573 | 35 | Vishniacozyma victoriae | Pathotroph-Saprotroph-Symbiotroph | Conventional |
| KY430574 | 17 | Holtermanniella | – | Conventional |
| KY430575 | 20 | Microdochium | Pathotroph-Symbiotroph | Conventional |
| KY430576 | 20 | Microdochium | Pathotroph-Symbiotroph | Conventional |
| KY430577 | 20 | Microdochium | Pathotroph-Symbiotroph | Conventional |
| KY430578 | 24 | Leotiomycetes | – | Conventional |
| KY430579 | 1 | Malassezia | Pathotroph-Saprotroph | Conventional |
| KY430580 | 0 | Agaricomycetes | – | Conventional |
| KY430581 | 18 | *Fusarium nurragi* | Pathotroph-Saprotroph-Symbiotroph | Conventional |
| KY430582 | 0 | Agaricomycetes | – | Conventional |
| KY430583 | 0 | Agaricomycetes | – | Conventional |
| KY430584 | 16 | *Paraphoma radicina* | Pathotroph-Saprotroph | Organic |

**Note:**
The accession numbers of sequences from this study are given as well as the assigned molecular operational taxonomic unit (MOTU), taxonomic information and ecological guild. Further Information can also be found in Table S4.

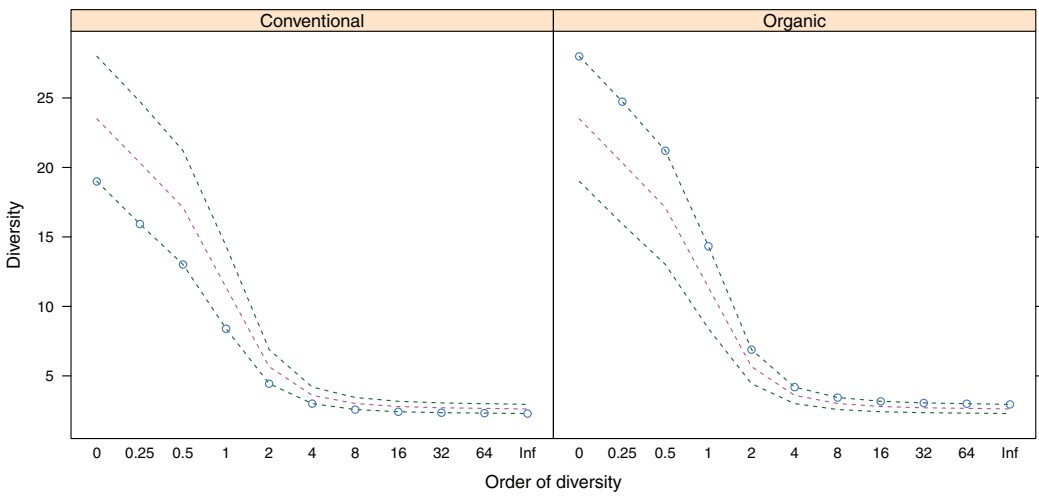

**Figure 3 Alpha-diversity (*y*-axis) of belowground fungal communities associated with roots of winter wheat (*Triticum aestivum*) from organic and conventional farming systems.** For each farming system, the Hill numbers for different values of *q* (order of diversity, *x*-axis) were calculated, giving different weights to species abundances. Green dotted lines show the minimum and maximum across all samples; pink dotted lines show the average.

## Belowground fungal community composition

Belowground fungal communities from both conventional and organic farming systems showed non-significant MPD values, whereas the MNTD values were significant for organic farming systems only ($p < 0.05$) (Table 1).
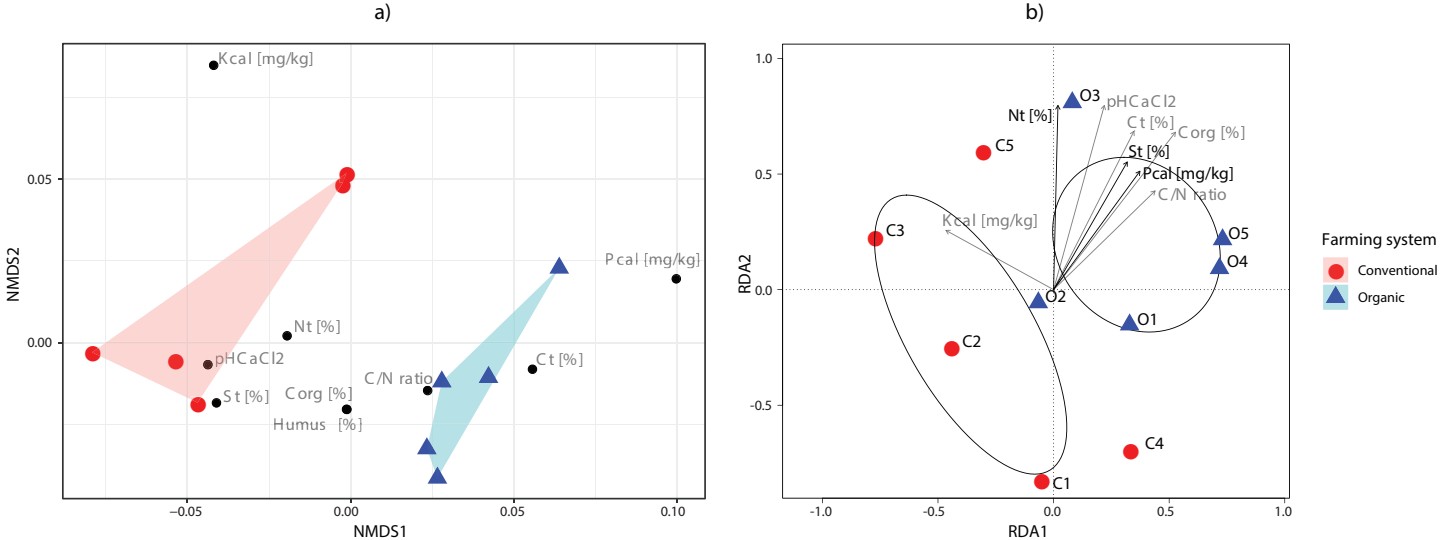

**Figure 4 Ordination analyses.** (A) Non-metric multidimensional scaling plot of sampling sites based on their environmental parameters. Polygons show the connexions of plots from organic (blue) and conventional (red) sites. (B) tb-RDA analysis of the fungal communities and the environmental variables associated with organic (blue) and conventional (red) sampling plots. Using variable selection, three environmental parameters (Nt, St, Pcal) were detected as having a significant effect on the fungal communities, and together accounted for 43.51% of the observed variation. Pcal: with Calcium-Acetate-Lactate extracted Phosphorus, Kcal: with Calcium-Acetate-Lactate extracted Potassium, Nt: Nitrogen total, Ct: carbon total, pH CaCl$_2$: soil pH, St: Sulfur total, Corg: Carbon organic, C/N: Carbon-to-Nitrogen ratio and Humus: humus content.

### Environmental factors of different farming systems

The main physical and chemical properties of the soils are shown in Table S3. The C/N ratios were higher in soils under organic farming practices than those under conventional practices. On the contrary, the related parameters total N, total C, organic C, phosphorus and humus contents had some values that overlapped between both management systems. Nonmetric multidimensional scaling (NMDS) showed the clear separation of farming systems according to the soil parameters analysed (Fig. 4A).

Variable selection with tb-RDA identified three significant variables that account for 43.51% of the variation (Ntotal ($p = 0.020$), Stotal ($p = 0.035$), Pcal ($p = 0.005$)) of the OTU distribution (Fig. 4B). The PERMANOVA, however, did identify the farming system as only having a minor effect, while not being significant ($p = 0.086$, $R^2 = 0.202$).

### Functional guilds

The fungal OTUs or species were grouped into 14 ecological guilds distributed across the following trophic modes: pathotrophs (two OTUs), saprotrophs (four OTUs), symbiotrophs (three OTUs), pathotroph-saprotrophs (six OTUs), pathotroph-symbiotrophs (four OTUs), pathotroph-saprotroph-symbiotrophs (four OTUs) and those with an unknown position in the food chain (16 OTUs) (Fig. 5; Table 1). In general, a higher diversity of trophic modes (involving some symbiotroph OTUs) were found on organically managed sites than in conventional farming systems (Table 1).

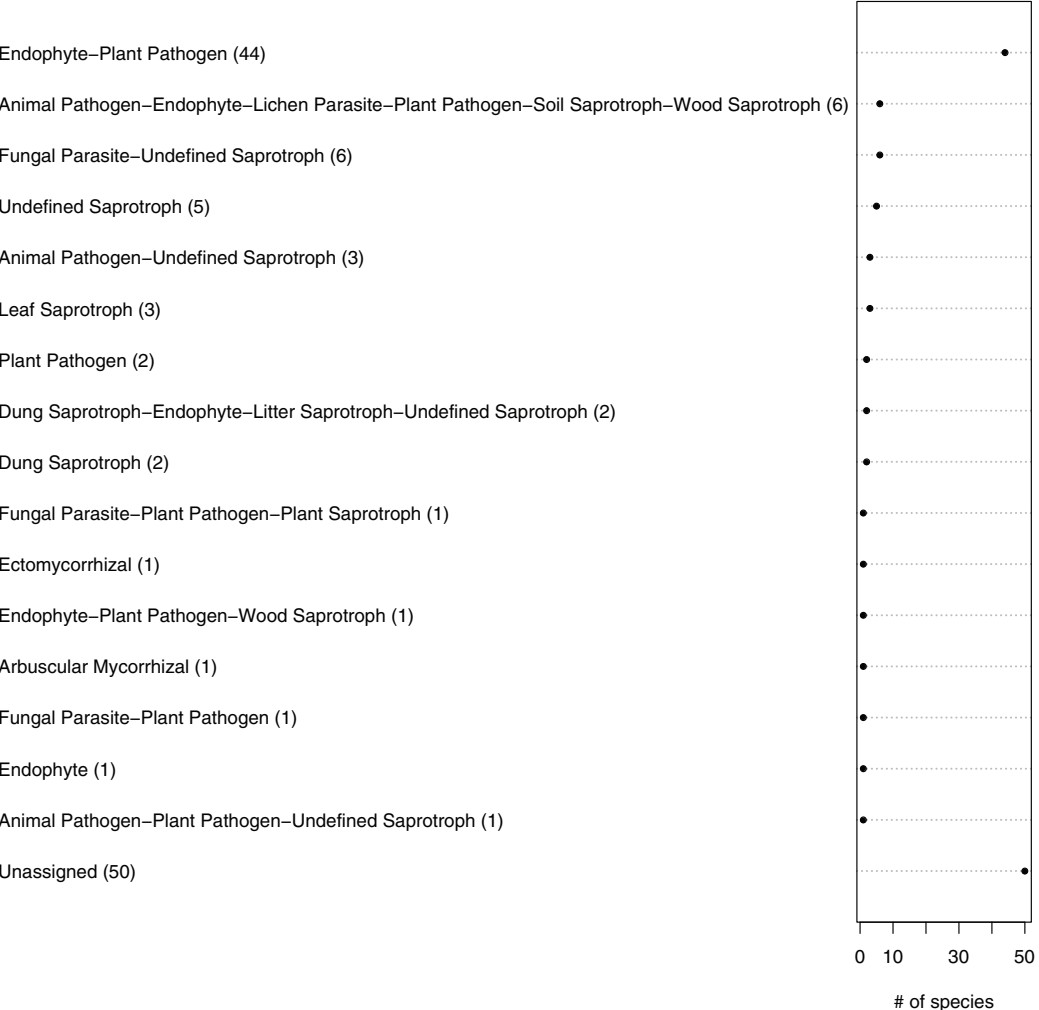

**Figure 5** **Ecological guilds of the fungal taxa in belowground communities associated with the roots of winter wheat (*Triticum aestivum*) from organic and conventional farming systems.** The fungal taxa were classified into 16 ecological guilds with the FUNGuild tool and database. The value in parentheses and the position on the *x*-axis indicate the number of individual sequences associated with that guild.

## DISCUSSION

### Taxonomic coverage and belowground fungal community diversity

The low number of PCR-positive samples obtained is possibly associated to the sample type rather than to the primers' specificity and sensitivity. We combined two sets of primers since only seven samples were positive with both sets. The remaining samples were positive with only one primer set. We suspect that factors such as the inadequate disintegration of fine roots or the presence of inhibiting substances from the host plant could have affected the detection success of fungi (PCR amplification) (*Schrader et al., 2012*).

In this study, in roots of *T. aestivum*, up to eight different OTUs were identified per sample, which indicates that the roots of one plant were colonised by several fungal species

simultaneously. *Verbruggen et al. (2014)* examined the fungal communities associated with wheat roots from organically and conventionally managed farming sites in Switzerland. They detected a remarkably greater number of OTUs (493) by sampling a smaller number of individual wheat plants from a larger number of sites. The higher diversity detected in their study may be partly caused by the different environmental conditions and sampling strategy, but was primarily caused by the molecular methodology used: they applied next-generation sequencing, in contrast to the Sanger sequencing we used here. Similar to this study, they identified most of the detected fungi as Ascomycota, with two of the most common OTUs clustered within the Sordariomycetes, and a few Basidiomycota (Fig. 2; Table 1). Similarly, *Carter et al. (1999)*, who isolated fungi from the roots of field-grown wheat and used cultivation methods and subsequent fungal characterisation by rDNA sequencing, found the Ascomycota to be more frequent than the Basidiomycota.

Although the organic farming sites and two of the conventional farming sites were located within short geographical distance from each other, each site had an almost unique OTU composition and the overlap of OTUs was low (Fig. 3). Conversely, some OTUs were frequently detected across several sampling sites, independent of the farming method. The OTU #20, identified as *Microdochium* sp., was more commonly detected in conventional than in organic sites. Members of the genus *Microdochium* are well known as plant pathogens which can cause head blight of cereals (*Xu et al., 2007*). The fact that this fungus was isolated from roots of healthy wheat plants confirms the common definition of endophytes, which also includes latent pathogens that at some stage of their lifecycle inhabit plant tissues without causing apparent harm to their host (*Petrini, 1991*; *Sun & Guo, 2012*). The OTU #7 (without any assignment beyond Ascomycota in the ITS RefSeq database) was closely related to *Lasiosphaeriaceae* spp. (sequences from GenBank database) and it has also been isolated from roots of *T. aestivum* in Sweden (*Grudzinska-Sterno et al., 2016*). *Lasiosphaeriaceae* spp. apparently include endophytic species with a broad host range; some members of this fungal family have also been isolated as endophytes of various host plants from diverse habitats (*Arnold et al., 2000*; *Su, Guo & Hyde, 2010*). In addition, OTU #26, which was identified as *Fusarium equiseti*, was detected in all conventional farming sites. *F. equiseti* is a ubiquitous soilborne pathogen that causes vascular wilt on a wide spectrum of plants (*Dean et al., 2012*).

## Impact of agricultural practices on belowground fungal communities

The present study revealed that the fungal communities are phylogenetically clustered and have significant NRI values, indicating that each farming method contains a distinctive fungal community (Table 2). Here, the farming method does influence the structure and also the fungal OTU diversity. In particular, these results highlight that fungal community assembly could result from similarities in response to some environmental factors, such as abiotic habitat filters, plant-imposed habitat filters or interactions with other fungal species (*Saunders, Glenn & Kohn, 2010*). Conventional farming systems, in contrast to organic systems, are characterised by the application of

**Table 2 Phylogenetic diversity and structure of the belowground fungal communities associated with roots of winter wheat (*Triticum aestivum*) collected from in conventional and organic farming systems.**

| Farming system | ntaxa | mntd.obs | mntd.rand.mean | mntd.rand.sd | mntd.obs.rank | mntd.obs.z | mntd.obs.p | Runs | |
|---|---|---|---|---|---|---|---|---|---|
| Conventional | 19 | 0.65646923 | 0.58565766 | 0.0390507 | 9,656 | 1.81332398 | **0.9656** | 9,999 | Dispersed |
| Organic | 28 | 0.52514116 | 0.56577169 | 0.02433867 | 478 | −1.6693818 | **0.0478** | 9,999 | Clustered |
| | ntaxa | mpd.obs | mpd.rand.mean | mpd.rand.sd | mpd.obs.rank | mpd.obs.z | mpd.obs.p | Runs | |
| Conventional | 19 | 0.82247012 | 0.80006999 | 0.02666562 | 7,886 | 0.84003779 | 0.7886 | 9,999 | |
| Organic | 28 | 0.78209976 | 0.79920574 | 0.01630606 | 1,539 | −1.0490569 | 0.1539 | 9,999 | |

Note:
The observed mean nearest phylogenetic taxon distance (MNTD) and mean phylogenetic distance (MPD), as well as the mean and standard deviation of 9,999 randomly generated communities, are given. The significance (*p*-values) of the MNTD and MPD values of samples was calculated with the 'picante' package in R (*Kembel et al., 2010*) by comparing the values with those of random communities.

agrochemicals, which can alter the soil's chemical composition and influence fungal diversity directly (*Ellouze et al., 2014*). *Verbruggen et al. (2014)* found that different agricultural management practices did not have a significant effect on the fungal communities associated with roots of *T. aestivum*, but they observed a significant effect of some abiotic soil factors like pH and magnesium concentration. The results for this analysis were however obtained by pooling sequences from different sites under the same management type, and further studies with an increased replicate number would be needed to verify the results obtained here. In this study, total N, total S and Pcal explain around half of the variation in abundance and composition of the fungal communities associated with winter wheat. Studying the long-term effects of nitrogen and phosphorus fertilisation on soil microbial community structure and function under continuous wheat production, *Li et al. (2019)* observed that N fertilisation increased fungal abundance. Similarly, *Jie et al. (2012)* found that sulphur affects the composition of arbuscular mycorrhizal fungal communities of different soybean cultivars. *Cai et al. (2019)* working with a rice-fish system found that total N and available P regulated the abundance of dominant fungi. When assessing the influence of the farming method directly, however, only a weak (and non-significant) effect could be observed, which could indicate that the underlying soil parameters play a stronger role than the type of management (REF).

One of the most striking differences between sampling sites from the two farming systems was the abundance and diversity of the co-occurring vegetation, which was generally higher in organic farming sites. As the OTU diversity was similarly high in organic sites, we postulate that the co-occurring vegetation could serve as a reservoir source of some fungal taxa. *Verbruggen et al. (2014)* examined the diversity of co-occurring weed communities and their ability to significantly influence fungal diversity and suggested that this might be partly a result of the weed litter left in the soil, which could stimulate the growth of endophytes with saprobic activity. Furthermore, they suggest the influence of the co-occurring plant community if some endophytic fungi with a broad host range proliferate more extensively in neighbouring plant species than in wheat itself. Besides that, there is evidence that plants release a wide array of compounds via their roots that can act as nutrient sources for soil fungi and can stimulate their proliferation (*Ellouze et al., 2014*). Therefore, a wide array of co-occurring plant species that release

equally diverse compounds may provide a basis for the establishment of a wide array of fungi. *Riess (2009)* studied endophytes associated with cereals, including *T. aestivum*, and the co-occurring herbaceous plant species and observed that in many cases, the neighbouring plants were colonised by the same fungal phylotype. This author suggested that the associated fungi could connect their host plants in belowground networks in a similar manner to ECM (*Onguene & Kuyper, 2002*) or arbuscular mycorrhizal fungi (*Voets et al., 2009*). This could also be supported by the fact that a great number of endophytic fungal species are primarily transmitted horizontally (*Rodriguez et al., 2009*).

The predicted functional guilds of the fungal sequences also varied between conventionally and organically grown winter wheat (Fig. 5; Table 1). Interestingly, an increase in the occurrence of potential plant pathogen fungi and a decrease in the symbiotrophs were predicted for organic farming. The fact that fungal orders containing mainly endophytic species were detected in the roots, as well as mycorrhizal, parasitic or saprobic species, is not surprising. For instance, members of the Glomeromycota, which comprise arbuscular mycorrhizal fungi and were detected twice in this study, can typically grow within plant roots (*Parniske, 2008*). Additionally, it is difficult to distinguish fungal species on the basis of their lifestyles, because some species change their ecological strategy during their lifecycle. For example there is evidence that some endophytes can also adopt saprobic lifestyles at senescence of their host plant (*Promputtha et al., 2007*; *Porras-Alfaro & Bayman, 2011*).

## CONCLUSIONS

In conclusion, our results show that the fungal communities associated with winter wheat are affected in their diversity, composition and functionality by the agricultural practices. We identify total N, total S and Pcal as having a significant effect on the OTU's abundance and distribution, but a large part of the variation within the studied communities remains unexplained by these factors. It seems likely that a correlation with the abundance and/or diversity of the co-occurring vegetation exists. Further studies that include a larger number of samples and apply integrated molecular and culture-based methods are needed. With this strategy, it would be possible to analyse the phylogenetic diversity and composition of the fungal communities more accurately in larger-scale comparisons of different faming systems. These studies should also address the role of the co-occurring vegetation in correlation with the fungal diversity and functionality associated with the roots of *T. aestivum*. Finally, the data obtained could be used to identify fungal species with beneficial effects on the fitness of *T. aestivum* and their potential to be used as organic fertilisers for agriculture.

### Funding

This research was financially supported by the German Research Foundation (DFG) Grant OB 24/30-1. The funders had no role in study design, data collection and analysis, decision to publish, or preparation of the manuscript.

## Grant Disclosures

The following grant information was disclosed by the authors:
German Research Foundation (DFG): OB 24/30-1.

## Competing Interests

The authors declare that they have no competing interests.

## Author Contributions

- Sigisfredo Garnica conceived and designed the experiments, analysed the data, prepared figures and/or tables, authored or reviewed drafts of the paper, and approved the final draft.
- Ronja Rosenstein performed the experiments, analysed the data, prepared figures and/or tables, authored or reviewed drafts of the paper, and approved the final draft.
- Max Emil Schön analysed the data, prepared figures and/or tables, authored or reviewed drafts of the paper, and approved the final draft.

## Field Study Permissions

The following information was supplied relating to field study approvals (i.e., approving body and any reference numbers):

The farmers Ulrich Bechtle (Unterjesingen) and Eckhart Wizemann (Entringen, Pfrondorf) allowed us to collect plant specimens from their crop fields.

## DNA Deposition

The following information was supplied regarding the deposition of DNA sequences:

Data is available at GenBank: KY430446–KY430584.

## Data Availability

The raw measurements are available in the Supplemental Files.

## Supplemental Information

Supplemental information for this article can be found online at http://dx.doi.org/10.7717/peerj.9732#supplemental-information.

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
