# Peer review of "Belowground fungal community diversity, composition and ecological functionality associated with winter wheat in conventional and organic agricultural systems"

_PeerJ, doi:10.7717/peerj.9732_

## Round 0.1 · original submission · Minor Revisions

Both reviewers and myself have seen merit in your work, originality and interest. However, some minor points and comments have been raised to improve its clarity and results. I invite you to consider all this comments and send a revised version of your manuscript.

Reviewer 1 ·

Basic reporting

The authors have explored the impact of two different agricultural practices on the mushroom community in winter wheat. The idea of preserving biological diversity is well established in both the plant and animal kingdom in densely populated environments.

This manuscript describes the results of the taxonomic classification of ITS genes isolated from roots and the ecological functionality of the fungal communities in this crop. Overall, this manuscript provides some useful ideas, but I have several comments/suggestions.

Experimental design

# 1
Illumina is nowadays the most used technology. If you need to identify the microbial diversity in a sample for sure you will get much more information using NGS techniques, after extracting DNA and amplifying ITS genes using different primers that are suitable for fungi. It would be very clear to the reader why the Sanger sequencing method was chosen instead of NSG in this project.ents.

Validity of the findings

I find a lack of statistical evidence to support the results. I have some suggestions:

# 1
In the section “Differences in environmental factors between farming methods” (lines 255-260), it is not mentioned the distance measure used for the distance-based ordination methods, in this case, NMDS. For example Euclidean, Bray or Jaccard distances among others.

#2
The most appropriate method to test the effect of exploratory variables on species abundance matrices is a CCA or RDA analysis followed by Variation Partitioning, in which we can see: What percentage of the total variance of the species distribution is capable of explaining each environmental factor and whether it is significant?
This type of analysis is much more complete and informative than the one shown in Figure 2.

In the following links, you can see how to perform this type of analysis in R.


Variable selection: https://www.davidzeleny.net/anadat-r/doku.php/en:forward_sel
Variation partitioning: https://www.davidzeleny.net/anadat-r/doku.php/en:varpart

#3
In order to statistically check if there are really differences between the different fungal communities according to the type of agricultural practice, I suggest performing a PERMANOVA test by means of distance-based dissimilarity matrices such as Jaccard (presence/absence of OTUs) or Bray-Curtis (abundance of OTUs).

Additional comments

# 1
Although the term mOTU is accepted and there are publications about it, I suggest changing it to OTU.

#2
In the results of the taxonomic classification, a total of 130 OTUs were classified at the phylum level. However, only 69% were classified at the order level (Figure 2). It would be interesting to know with your methodology what percentage of OTUs are classified up to the genus or species level and to see if these values improve using a much more extensive database. For instance, the RefSeq ITS (https://www.ncbi.nlm.nih.gov/bioproject/177353) consists of 11, 250 entries, while the database "UNITE + INSD" (https://unite.ut.ee/repository.php) consists of 714, 329 for the ITS gene on fungi.

Reviewer 2 ·

Basic reporting

NB: this review was prepared without explicit consideration of the different "review paragraphs" - therefore it only loosely follows this structure.

The manuscript entitled "Belowground fungal community diversity, composition and ecological functionality associated with winter wheat in conventional and organic agricultural systems" by Garnica et al. describe a phenomenon of general scientific interest, which is the diversity and community composition of fungi that associate with roots of crops under different agricultural management settings. This relevance is clearly set out in the introduction, and further elaborated on in the discussion, in a generally well-written manuscript.

Experimental design

After raising high expectations in the abstract, it turns out that the obtained results are quite limited in extent: L42 suggest results of roots from 500 plants will be presented, but in fact they are only 100 "pooled" samples. Moreover, from the results it is not quite clear exactly which of these actually contributed to the results (it is only apparent after checking Table S2, but should already be clear there). In line 272 it is said that "Out of the 100 pooled root samples, 54 pooled root samples yielded PCR products either directly or after cloning" – this makes me immediately wonder: how can you clone an amplicon without PCR product? This confusion remains, in the line immediately after the numbers don't add up (how do 15 direct sequences and 26 cloned sequences (sum=41) represent 43 samples? Same in the next sentence. These are the first lines of the results which is where readers are going to get confused and stop reading I'm afraid.

Validity of the findings

From this point onwards the fact that plants originated from 5 different fields is entirely ignored. This is never explicitly justified by the authors, but is presumably the case because so few samples gave rise to usable sequences. This is something that could be reflected on more extensively: why did so few samples produce PCR products? Was there some sort of inhibition problem? Nevertheless, just treating samples as coming from either organic or conventional agriculture causes replication to be essentially N=1. This makes it difficult to claim that particular functional groups were overrepresented in e.g. conventional agriculture, or that one was more phylogenetically clustered than the other. I am sympathetic to authors that they are trying to see light in limited data, and they are generally quite careful in phrasing conclusions, but I feel it should be remarked upon.

Additional comments

L48 (Abstract): here the claim is made that "fungal communities from conventional farming sites were significantly clustered" – without further context in the abstract this is difficult to interpret.
L86: Here it is suggested that Verbruggen et al. (2014) did not find differences in Sebacinales occurrence between land-use types, but in fact they claimed there were.

There are font-size changes in both M&M, Results, and Discussion sections.

L317: "In general, a higher number of different trophic modes (involving some symbiotroph
MOTUs) were found in organically managed sites than in conventional farming systems." – this is an ambiguous statement; do you mean they are more equally distributed according to different trophic modes?
L397: "Predicted of functional groups" – please rephrase
L410: "the intrinsic nature of environmental factors" – vague statement
L411: "it" seems likely

Figure 1: While there are 10 sites that have been sampled, only six are shown in Figure 1? The caption is very unclear and should be revised.
Figure 2: here there are some misspellings: "Magnaporthales", "Helotiales" – especially t's seem to be conspicuously missing?

---

## Round 0.2 · accepted · Accept

Thanks for reviewing your manuscript deeply and taking into account all comments raised by both reviewers. I believe that your manuscript has improved after the review process. Congratulations!

Reviewer 1 ·

Basic reporting

no comment

Experimental design

no comment

Validity of the findings

The section "Environmental factors of different farming systems" reads a lot better and the authors have addressed most of my comments.

The results obtained by RDA and PERMANOVA tests provide further statistical support to the manuscript.

Additional comments

Authors answers to all the question and meliorate the manuscript.

Reviewer 2 ·

Basic reporting

no comment

Experimental design

no comment

Validity of the findings

no comment

Additional comments

I hereby re-review the paper I commented on earlier. I believe the authors have taken all objections raised by me and the other referee seriously and have incorporated suggestions adequately. I have no further objections and congratulate the authors on their nice work.